# Tissue Specific Distribution and Activation of *Sapindaceae* Toxins in Horses Suffering from Atypical Myopathy

**DOI:** 10.3390/ani13152410

**Published:** 2023-07-26

**Authors:** Johannes Sander, Michael Terhardt, Nils Janzen, Benoît Renaud, Caroline-Julia Kruse, Anne-Christine François, Clovis P. Wouters, François Boemer, Dominique-Marie Votion

**Affiliations:** 1Screening-Labor Hannover, 30952 Ronnenberg, Germany; m.terhardt@metabscreen.de (M.T.); n.janzen@metabscreen.de (N.J.); 2Department of Clinical Chemistry, Hanover Medical School, 30625 Hanover, Germany; 3Department of Functional Sciences, Faculty of Veterinary Medicine, Pharmacology and Toxicology, Fundamental and Applied Research for Animals & Health (FARAH), University of Liège, 4000 Liège, Belgium; benoit.renaud@uliege.be (B.R.); acfrancois@uliege.be (A.-C.F.); clovis.wouters@uliege.be (C.P.W.); dominique.votion@uliege.be (D.-M.V.); 4Department of Functional Sciences, Physiology and Sport Medicine, Faculty of Veterinary Medicine, Fundamental and Applied Research for Animals & Health (FARAH), University of Liège, 4000 Liège, Belgium; caroline.kruse@uliege.be; 5Biochemical Genetics Laboratory, Human Genetics, CHU Sart Tilman, University of Liège, 4000 Liège, Belgium; f.boemer@chuliege.be

**Keywords:** equine atypical myopathy, hypoglycin A, methylenecyclopropylacetate, *Sapindaceae* toxin, acylcarnitines

## Abstract

**Simple Summary:**

Equids kept at pasture are at risk of being intoxicated by ingesting sycamore maple seeds or seedlings, which contain the two non-proteinogenic amino acids hypoglycin A and methylenecyclopropylglycine. These amino acids are converted into effective toxins by metabolic processes, inducing severe damage to oxidative muscles. Toxic metabolites are known to disrupt the cellular use of important energy sources such as short- and medium-chain fatty acids or branched-chain amino acids. The comparative examination of different tissues from five horses that died from this environmental intoxication named atypical myopathy revealed that the highest concentration of active toxins was found in muscles. In all the tissues analyzed, there was still unmetabolized hypoglycin A, which suggests that inhibiting the conversion of protoxins into toxic metabolites would be a possible therapeutic approach.

**Abstract:**

Equine atypical myopathy is caused by hypoglycin A (HGA) and methylenecyclopropylglycine (MCPrG), the known protoxins of sycamore maple (*Acer pseudoplatanus*). Various tissues from five atypical myopathy cases were analyzed but only HGA was found. Whether deamination of MCPrG has already occurred in the intestine as the first stage of metabolization has not been investigated. Activation of the protoxins to methylenecyclopropylacetyl (MCPA)-CoA and methylenecyclopropylformyl (MCPF)-CoA, respectively, occurred mainly in the skeletal muscles, as evidenced by very high concentrations of MCPA-carnitine and MCPF-carnitine in this tissue. Inhibition of the acyl-CoA dehydrogenases of short- and medium-chain as well as branched-chain fatty acids by the toxins led to a strong increase in the corresponding acylcarnitines, again preferentially in skeletal muscles. An accumulation of the long-chain acylcarnitines beyond the level of the control samples could not be detected in the tissues. As a high amount of HGA was always found unmetabolized in the organs, we speculate that targeting the interruption of further metabolization might be a way to stop the progression of intoxication. Inhibition of the mitochondrial branched-chain amino acid aminotransferase, i.e., the first enzyme responsible for the activation of sycamore maple protoxins, could be a therapeutic approach.

## 1. Introduction

Equids grazing on European pasture are at risk of lethal intoxication by ingesting seeds and seedlings from the sycamore maple (*Acer pseudoplatanus*) [1,2,3,4,5,6]. Botanically, maple trees belong to the *Sapindaceae* family. Deaths caused by components of these trees have also been described in zoo animals [7,8,9,10]. Intoxications by *Sapindaceae* toxins are also of concern for human medicine [11,12,13,14]. The proven toxic constituents are hypoglycin A (methylenecyclopropylalanine, HGA) and methylenecyclopropylglycine (MCPrG). Both also occur as dipeptides conjugated with glutamic acid (hypoglycin B and α-glutamyl MCPrG, respectively) [15]. Both HGA and MCPrG are not toxic by themselves. They are rather to be regarded as protoxins that have to be transformed into toxically active compounds.

The metabolism of HGA and MCPrG is known to duplicate the first two steps occurring with branched-chain amino acids [16,17]. The enzymatic activation takes place after the so-called LAT1 transporter, an ubiquitously expressed transport protein from the system L transporter family, has effected the transfer into the intracellular space. The first reaction there consists of the deamination of HGA or MCPrG by a branched-chain aminotransferase (BCAT, EC 2.6.1.42) [18,19]. Thus, MCP-pyruvate is formed from HGA and MCP-glyoxalate from MCPrG.

Unlike other amino acids, the first step of catabolism of the branched molecules does not take place preferentially in the liver but in the skeletal muscles. This gives rise to the assumption that the deamination of HGA and MCPrG can also take place to a large extent in this tissue. One of the aims of our study was to test this theory.

In a second enzymatic step, the α-keto acids formed by transamination are converted by the branched-chain α-keto acid dehydrogenase complex (BCKDHc; EC 1.2.4.4) into the corresponding acyl-CoA derivatives. The activity of the enzymes varies from tissue to tissue, both inter- and intraspecifically, and can also show large differences within an organ [20,21,22]. The products of the toxin activation are the CoA esters of methylenecyclopropylacetate (MCPA) and methylenecyclopropylformate (MCPF), respectively. Regarding the tissue-specific enzyme activities, we wanted to examine to what extent an activation of the protoxins shows a tissue-typical pattern.

The clinical presentation of *Sapindaceae* intoxication is characterized by acute onset and high lethality. Death usually occurs within 2 days after the onset of the clinical manifestation. An acute rhabdomyolysis syndrome combined with marked hyperglycemia determines the clinical picture in equids [23,24]. In horses, initial knowledge of the pathophysiological process has improved supportive therapy [25] and refined the decision criteria for euthanasia [26].

Necroscopic examination reveals extensive muscle necrosis, predominantly in the postural and respiratory muscles. The extent and severity of the changes vary considerably, not only between individuals but also between the different muscles of each animal. It is worth noting that in some cases, macroscopic lesions are not observed. Within an affected muscle, the degree of myodegeneration may vary greatly. Histology revealed abundant neutral fat, predominantly in type 1 muscle fibers [27,28]. Electron microscopy shows morphological changes in the mitochondria [27,29]. Westermann et al. found deficiencies of short-chain and medium-chain as well as isovaleryl-CoA dehydrogenases in muscle tissue [29]. The plasma/serum acylcarnitines profile showed increased concentrations of short- and medium-chain carnitine esters [29] but also of long-chain acylcarnitines [26]. A further aim of our investigations was therefore to examine whether the expected tissue-specific activation of the protoxins would also lead to a tissue-specific accumulation of products of the energy metabolism.

In some horses, severe macroscopic and histological changes in the myocardium have been found with the presence of pale areas, accumulation of neutral fat, and/or granular degeneration of myocardium cells, while no lesions were found in others [6,23,27,28]. However, elevated troponin I in the blood [26] and both electrocardiographic and echocardiographic changes were seen in most investigated cases [30]. Inhibition of 2-ketoglutarate and pyruvate dehydrogenases in bovine heart mitochondria [31] and reversible inhibition of long-chain acyl-CoA:carnitine acyltransferase were described [32].

Visual examination of other equine tissues such as the central and peripheral nervous systems or the pancreas did not reveal specific lesions; however, forebrain swelling with cellular oedema was reported as well as some little hemorrhage foci on the meninges in the first clinical series. Apart from hyaline, granular, or myoglobin-containing cylinders in the tubules of the kidney, the parenchymatous organs showed no histological abnormalities [27,28]. In rats, ultrastructural electron microscopic examination revealed swelling of the mitochondria of liver cells and a reduction in matrix density 3–5 h after intraperitoneal injection of a high HGA dose (i.e., 100 mg/kg; [33]).

In the case of poisoning, routine laboratory tests show a severe increase in muscle enzyme activity and, most of the time, hyperglycemia, high haptoglobin concentration, hyperlipaemia, high troponin I, and increased liver enzyme activity [23,24,26]. In the urine of horses, myoglobin is excreted in large quantities [24,29]. Such analyses are valuable diagnostic tools in cases of *Sapindaceae* intoxication. However, they do not provide any information about the affected tissues.

The therapy of maple intoxication has so far only been successful to a very limited extent. If one wishes to improve it decisively, more precise knowledge of the pathobiochemistry is required. This study is intended to make a contribution to this.

## 2. Materials and Methods

### 2.1. Horses

Five horses with a tentative diagnosis of atypical myopathy according to a diagnostic algorithm [34] used in the literature [26,35,36] were included in this study (Table 1). Serum samples were obtained from four of them just prior to euthanasia being decided due to the progressively worsening of respiratory difficulties [37]. No blood sample was available for one horse that was found dead in pasture. All five horses were necropsied between 1 and 4 days after spontaneous death or euthanasia.

Samples of the *semitendinosus*, *triceps brachii*, *gluteus medius*, myocardium, and diaphragm muscles were taken, as well as samples of the liver, kidney, and pancreas. The cerebrospinal fluid was collected from three horses during necropsy. In addition, in the four euthanized animals, collection of samples of the *semitendinosus* muscle took place not only at necropsy but also immediately after death.

Tissues from five slaughtered animals served as controls. The pancreas and cerebrospinal fluid of unaffected horses were not available. An undefined number of days elapsed between slaughtering and the acquisition of the material, during which the samples had been kept refrigerated but not frozen. Serum samples for control were taken from material examined for diseases not related to maple intoxication. Once collected, all samples were stored frozen at −18 °C until analysis.

### 2.2. Preparation and Use of Tissue Extracts

Quantities of 30 to 130 mg of material were separated from the frozen tissue samples using a scalpel, and 1000 µL of methanol was added. After brief mixing, samples were placed in an ultrasonic bath for 40 min at room temperature. Centrifugation at 14,000× *g* for 10 min produced a clear supernatant. A dilution factor was calculated from the ratio of the tissue weight to the amount of methanol. Analyses of liquid materials were conducted on 30 µL each.

It is not possible to give a completely accurate indication of the concentrations of protoxins, toxins, their metabolites, or acyl compounds because it is not possible to load animal tissues with these compounds in a reproducible manner. However, repeated extractions can give an indication of the degree of completeness of the initial extraction. It was found that only about 10% of the original concentration is obtained with a second extraction. Therefore, we assume that our analytical values correspond to about 90% of the total tissue contents.

### 2.3. Quantification of Toxins, Toxin Metabolites, and Several Acyl Metabolites

Ultra-performance liquid chromatography coupled with tandem mass spectrometry using a Xevo TQMS (Waters, Eschborn, Germany) was used for the quantification of HGA and MCPrG plus their metabolites in serum and tissue extracts, as described in detail earlier [38,39,40,41]. In addition, concentrations of C4 to C10 acyl conjugates were determined. This method also allowed for the differentiation of branched and unbranched C4 and C5 metabolites. Further carnitines and glycines were determined comparatively in four tissues of one horse. Statistical data on the normal distribution of concentrations of acyl compounds in the tissues of healthy horses are not available. Therefore, in this study, mean values obtained by examination of tissues from five healthy slaughtered animals were used for comparison.

Analyses were performed after butylation. The butyl esters were detected in electrospray ionization-positive mode by multiple reaction monitoring. For the separation of analytes, 5 µL of the final extracts were injected into an Acquity UPLC BEH C18 1.7 µm, 2.1 *×* 50 mm column (Waters, Eschborn, Germany). For gradient chromatography, acetonitrile/water eluents modified by 0.1% formic acid and 0.01% trifluoroacetic acid were used.

## 3. Results

### 3.1. Toxins and Toxin Metabolites

Neither toxins nor toxin metabolites of *Sapindaceae* were found in any of the control samples. As shown in Table 2, non-metabolized HGA was detected in all tissues as well as in the cerebrospinal fluid and the serum of the patients. Unmetabolized MCPrG (not listed in Table 2) was detected in trace amounts only. Concentrations of HGA differed based on the type of material analyzed. There were also large individual differences, as can be derived from the wide range of measured values. In skeletal muscles, including the diaphragm, HGA concentrations were generally found to be lower than in the heart, liver, kidney, and serum. The concentrations were particularly high in the pancreas. Concentrations here exceeded those of *M. semitendinosus* on the same horse by factors up to 23.8.

The distribution of the carnitine and glycine conjugates of MCPA and MCPF showed a pattern very different from that of HGA. MCPA-carinitine, formed from MCPA-CoA by conjugation with carnitine, as well as MCPF-carnitine, formed from MCPF-CoA, were present in very high concentrations in skeletal muscles. In contrast, in the cardiac muscle, the level of MCPA-carnitine was on average only 1.3% of the simultaneously present HGA. In the parenchymal tissues too, the metabolization of HGA was lower than in the skeletal muscles. Accordingly, the concentration of HGA was markedly higher there. The measured values for MCPA-carnitine in these organs, with a few exceptions in liver and kidney samples, were considerably lower than 10% of those of HGA.

MCPA-CoA and MCPF-CoA were conjugated with carnitine or glycine to quite different extents. In skeletal muscles, conjugation with carnitine was by far predominant for both CoA compounds. Again, this was not true for the cardiac muscle. Comparatively strong conjugation with glycine was found in the liver and was particularly pronounced in kidney tissue, which is known to have a high activity of glycine acyltransferase.

Results of a study on the post-mortem stability of HGA and MCPA-carnitine as well as three acylcarnitines in *M. semitendinosus* are summarized in Table 3. There was only a slight decrease in the concentration of HGA but a significant reduction in the concentration of MCPA-carnitine. For the acylcarnitines examined, there was a decrease in concentration by factors of up to 6.3.

### 3.2. Acylcarnitines and Acylglycines

As with toxin metabolites, tissue-specific accumulations of acyl compounds have been observed. Again, the skeletal muscles were particularly affected, as shown in Table 4.

In the four serum samples collected from 4 of the 5 horses with atypical myopathy, an increase in all acylcarnitines and hexanoylglycine was observed compared to samples collected from horses not affected by sycamore maple intoxication. In the tissues, however, this was only partly the case. Severely increased values were found for short- to medium-chain compounds in the skeletal muscles, but with large individual differences. The very high values for butyrylcarnitine prove a strong inhibition of the enzyme short-chain acyl-CoA dehydrogenase (EC 1.3.99.2). As an indication of the inhibition of the ß-oxidation of medium-chain fatty acids and the degradation of branched amino acids, the corresponding carnitine conjugates were found in the skeletal muscles of the diseased horses in considerably elevated concentrations. Additionally, the simultaneous accumulation of the monounsaturated decenoylcarnitine (C10:1), which is typical for cases of medium-chain acyl-CoA dehydrogenase (EC 1.3.8.7) deficiency in humans, was also observed here.

In the myocardium, accumulation of straight-chain acylcarnitines was only observed in two patients. In four cases, increased concentrations of the branched compounds isovalerylcarnitine and 2-methylbutyrylcarnitine were detected, but the maximum concentrations were often more than one order of magnitude lower than those in the skeletal muscle of the same animals.

In the liver, kidney, and pancreas, control levels were strongly exceeded only in some cases; in numerous samples, the measured values were in the control range.

Looking at the long-chain acylcarnitines, we found an enrichment of myristoyl-, palmitoyl-, and stearylcarnitine as well as of oleylcarnitine in the serum. Although higher concentrations of long-chain acylcarnitines were found in the skeletal muscles than in other tissues, similar levels were also observed in some controls. Thus, a toxin-dependent accumulation of long-chain acylcarnitines in these tissues was not proven.

The conjugation of the acyl residues in the skeletal muscles was predominantly with carnitine, while binding to glycine occurred to a much lesser extent. It is also evident that the ratios in the heart muscle and in the other tissues diverge strongly from this.

Table 5 illustrates the strong predominance of conjugation with carnitine versus glycine not only for butyric acid but also for branched fatty acids additionally quantified in the tissues of one horse.

## 4. Discussion

This paper is the first to show that HGA is distributed unhindered to all organs after intestinal absorption and that the protoxins are activated to a very unequal extent in the different organs to form the toxic products. The tissue-specific, varying accumulation of products of the energy metabolism was also demonstrated here for the first time. For the assessment of the quantitative results presented here, however, certain limitations should be considered.

Some uncertainty results from the fact that control tissues from slaughtered horses had to be used. The blood is always drained when the animal is for human consumption, but this is not the case when the animal is kept for necropsy. Tissue subtypes such as certain muscle fibers or the renal medulla and cortex could not be examined separately. Furthermore, it must be considered that the collection of material at any point in time always generates an instantaneous value at that very point in time. Pre-mortem changes may take place, and post-mortem alterations have been shown in this study. In addition, since horses on pastures ingest sycamore maple materials irregularly, quite different phases of resorption, metabolization, and excretion can be encountered at the time of the pathological-anatomical examination. HGA is known to be rapidly absorbed from the intestine. However, subsequent metabolization then takes place over a longer period [40,41]. There must be an accumulation when the toxin is taken up again in the decay phase.

Despite the uncertainties described, some basic statements can be made: one of the most important observations is that there is an unhindered distribution of HGA to different tissues, corresponding to an unmetabolized passage of the physiologic branched amino acids through the liver. The concentrations of HGA, however, were low in the cerebrospinal fluid. In the three cases examined, levels of only 212 to 356 nmol/L were found. Studies on a specific blood/brain barrier have not yet been published.

The metabolization rates of the protoxins proved to be very different in a tissue-typical manner. The very large differences in the concentrations of MCPA and MCPF derivatives found in the different tissues largely rule out an effective exchange of these metabolites between organs. It is also shown that the accumulation of the metabolites coming from the β-oxidation of fatty acids under the influence of the *Sapindaceae* toxins obeys a pattern typical of each tissue.

So far, little attention has been paid to the fact that sycamore maple materials contain not only HGA but also MCPrG [15,42], which, however, was not identified in significant concentrations in the serum or in the tissues in our investigation. The reason behind this is unknown. Whether instability in the matrix or rapid metabolization could be responsible requires further investigation. If MCPrG is absorbed in bound form, e.g., as a dipeptide with glutamic acid, it would not be detected with the analytics used in this study. In any case, the high concentrations of MCPF derivatives in skeletal muscle and other tissues suggest that MCPrG is a major contributor to the clinical manifestations of sycamore maple poisoning in equids.

In skeletal muscles, the CoA metabolites of HGA and MCPrG were preferentially conjugated with carnitine (Table 2). The use of carnitine in the treatment of HGA poisoning has occasionally been discussed [25,43]. The very high concentration of carnitine esters in the muscles, even without carnitine application, calls for caution. In contrast, in the liver and especially in the kidney, conjugation of MCPA but not MCPF also occurred to a greater extent with glycine. While MCPA carnitine was only released into the blood to a small extent, the glycine conjugate was detectable in the serum in slightly higher concentrations. The opposite was true for MCPF. An explanation of the tissue- and compound-specific differences would require further investigation.

It should be pointed out that at the time of euthanasia or spontaneous death, HGA was found in considerable quantities in the tissues and blood in an unmetabolized state. Large amounts of not yet metabolized HGA must be expected, especially before the clinical signs are approaching their peak. If one wants to intervene early, it is crucial to prevent further activation of HGA by MCPA-CoA. As HGA activation requires the interaction of several proteins, interruption of the function of one of these might be therapeutically useful. The importance of BCAT for the activation of sycamore maple toxins could be of interest because an inhibitor of the enzyme, gabapentin, is already being used in another veterinary context [44]. Furthermore, previous studies had shown that in the stomach and intestine, HGA can still be found in undigested sycamore maple material and is unabsorbed in the intestinal lumen [8,9]. Gastric/intestinal lavage and the application of activated charcoal could prevent further absorption to a certain extent [45].

## 5. Conclusions

In addition to HGA testing, a profound diagnosis of atypical myopathy additionally requires the quantitative determination of toxin metabolites as well as evidence of the interrupted ß-oxidation of fatty acids and of the disturbed energetic utilization of the branched amino acids. The subsequent severe skeletal muscle damage results in a sharp increase in serum creatine kinase activity, which confirms the rhabdomyolysis syndrome.

In light of the findings of this study, new therapeutic approaches could be considered, in particular the inhibition of the enzymes and transport systems that allow protoxins to become effective toxins. These possibilities could be used both for prevention (the toxins being ubiquitous in certain regions) and for emergency treatment, as the protoxins are still present in the blood and tissues of poisoned animals.

## Figures and Tables

**Table 1 animals-13-02410-t001:** Characteristics of atypical myopathy-affected horses.

Horse	Breed	Sex	Age (in Years)	Date of First Signs	Date of Death	Cause of Death
1	Quarter-horse	Female	>24	unknow	3 November 2020	Found dead
2	Spanish Purebred	Stallion	7	31 October 2020	3 November 2020	Euthanasia
3	Belgian Draft Horse	Stallion	3.5	31 October 2020	1 November 2020	Euthanasia
4	Half-blood	Stallion	1.5	27 October 2020	31 October 2020	Euthanasia
5	Percheron	Gelding	4	24 October 2020	26 October 2020	Euthanasia

**Table 2 animals-13-02410-t002:** Hypoglycin A, its metabolites (MCPA-carnitine and MCPA-glycine), and metabolites of methylenecyclopropylglycine (MCPF-carnitine and MCPF-glycine) in different tissues and cerebral fluid collected at necropsy as well as in serum collected prior to euthanasia. The concentrations are expressed in nmol/kg in tissues and in nmol/L in the fluid samples.

Organ	Concentration	HGA	MCPA-Carnitine	MCPA-Glycine	MCPF-Carnitine	MCPF-Glycine
*M*. *semitendinosus*.	Mean	496	2078	35	7710	20
	Range	103–1379	156–3690	6.5–47	539–23,416	0.8–43
*M*. *triceps brachii*.	Mean	536	1884	43	7260	22
	Range	111–1473	86–4108	1.3–75	10–16,100	0.1–38
*M*. *gluteus medius*	Mean	528	2883	70	9595	35
	Range	195–1259	40–7482	16–109	78–25,782	5.1–80
Diaphragm	Mean	606	639	123	4824	33
	Range	307–1360	5–1431	14–318	6.9–19,056	3.3–83
Myocardium	Mean	1115	15	99	27	33
	Range	346–3527	0.7–54	13–165	2.1–109	6.8–47
Liver	Mean	1340	98	244	240	109
	Range	606–3578	0.8–460	39–542	2.6–1142	14–307
Kidney	Mean	775	71	1655	696	309
	Range	198–2029	3.0–208	96–3231	3.3–2495	35–629
Pancreas	Mean	2209	75	70	246	33
	Range	184–5237	1.6–162	21–145	14–795	21–55
Serum	Mean	1831	50	253	716	91
	Range	909–2948	36–64	154–296	541–958	54–114
Cerebral. fluid	Mean	269	42	50	398	23
	Range	213–356	2–61	<1–67	26–503	<1–26

Abbreviations: HGA—Hypoglycin A; MCPA—methylenecyclopropylacetyl; MCPF—methylenecyclopropylformyl.

**Table 3 animals-13-02410-t003:** Reduction factor for concentrations measured in samples of *M. semitendinosus* collected immediately after death and collected at necropsy.

Horse	Days between Death and Necropsy	HGA	MCPA-Carnitine	Butyryl-Carnitine	Hexanoyl-Carnitine	Octanoyl-Carnitine
2	2	1.1	2.9	2.6	5.1	6.3
3	2	1.2	2.1	1.9	2.1	3.5
4	3	1.2	2.6	1.2	1.9	2.1
5	1	1.1	2.3	1.9	4.4	4.0

Abbreviations: HGA—hypoglycin A; MCPA—methylenecyclopropylacetyl.

**Table 4 animals-13-02410-t004:** Tissue-specific concentrations of straight-chain and branched-chain acyl compounds (in µmol/kg) in (**A**) atypical myopathy-affected horses and (**B**) controls.

(A)									
Organ	Concentration	C4-C	C6-C	C6-G	C8-C	C10:1-C	Isobutyryl-C	Isovaleryl-C	2-MB-C
*M. semitendinosus*.	Mean	1464	66	0.6	14	2.2	49	515	225
	Range	157–3548	13–201	0.1–0.9	2.9–42	0.2–6.3	22–92	12–1010	16–480
*M. triceps brachii*.	Mean	1232	68	0.7	13	2	52	401	166
	Range	1.8–2241	8.9–130	0.5–1.2	1.1–33	0.2–4.1	<0.1–114	22–703	47–450
*M. gluteus medius*	Mean	1479	70	1.1	18	6.8	45	489	233
	Range	38–4369	7.3–169	0.2–2.1	0.6–55	<0.1–28	0.2–123	19–1103	22–430
Diaphragm	Mean	1203	47	1.3	6.2	0.3	23	260	78
	Range	0.7–3920	2.7–107	0.2–2.4	0.1–10	<0.1–0.5	<0.1–65	8.7–653	17–218
Myocardium	Mean	29	0.8	1	0.3	-	0.4	8.9	6.6
	Range	3.1–54	0.1–2.9	0.1–1.8	<0.1–0.5		<0.1–0.8	0.4–18	1.9–9.4
Liver	Mean	139	4.7	2.2	1.4	0.4	1.7	33	7.9
	Range	0.7–680	0.2–21	1.1–4.4	<0.1–5	<0.1–0.6	<0.1–5.3	0.8–130	<0.1–28
Kidney	Mean	119	5.3	6.3	0.6	0.4	4.1	43	13
	Range	1.9–492	0.3–20	0.4–18	0.1–2.1	<0.1–0.5	<0.1–8.1	2.1–123	3–36
Pancreas	Mean	47	4.1	1	1	0.2	1.2	20	6.3
	Range	3–149	0.2–9.3	0.1–1.3	0.1–3	<0.1–0.3	0.3–2.6	3.3–43	0.7–9.8
Serum	Mean	41	3.6	4.4	0.8	0.6	2.9	13	6.2
	Range	27–58	2.8–4.4	2.2–6.1	0.7–1	0.4–0.6	1.2–4.3	9.9–20	3.4–8.3
(**B**) unaffected controls (means of 5 samples each)									
**Organ**	**Concentration**	**C4-C**	**C6-C**	**C6-G**	**C8-C**	**C10:1-C**	**Isobutyryl-C**	**Isovaleryl-C**	**2-MB-C**
Skeletal muscles	Mean	0.2	0.5	0.1	<0.1	<0.1	<0.1	2.8	11.4
Myocardium	Mean	8.2	0.7	-	0.1	<0.1	0.4	0.4	1
Liver	Mean	0.9	0.4	0.3	<0.1	<0.1	1.7	26	5.3
Kidney	Mean	2.6	2.3	0.1	0.1	<0.1	0.7	0.5	0.6
Serum	Mean	0.2	<0.1	<0.1	<0.1	<0.1	0.9	0.1	0.2

Abbreviations: C4-C—butyryl carnitine; C6-C—hexanoyl carnitine; C6-G—hexanoyl glycine; C10:1-C—decenoyl carnitine; 2-MB-C—2-methylbutyryl carnitine; -C—carnitine.

**Table 5 animals-13-02410-t005:** Ratio of acylcarnitines per acylglycines in tissues of different organs of horse 1 sampled at necropsy.

Organ	C4-C /C4-G	Isobutyryl-C /Isobutyryl-G	2-MB-C /2-MB-G	Isovaleryl-C /Isovaleryl-G
*M. semitendinosus*	2343	9383	20262	4320
Liver	183	144	82	121
Kidney	10	55	17	9
Myocardium	14	40	68	22

Abbreviations: C4-C or -G—butyryl carnitine or glycine; 2-MB-C or -G—2-methylbutyryl carnitine or glycine.

## Data Availability

The data presented in this study are available upon request from the corresponding authors.

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
