# Peer review of "Tissue Specific Distribution and Activation of Sapindaceae Toxins in Horses Suffering from Atypical Myopathy"

_animals, 2023, doi:10.3390/ani13152410_

Round 1

Reviewer 1 Report

I am pleased to have the opportunity to review a paper under the title  “Tissue specific distribution and activation of Sapindaceae toxins in horses suffering from atypical myopathy”.
As a result of the review, section material and methods with results has to be reorganized for better understatement of the number of animals used within the study. This is not clear how many horses and tissues were used. How many cases, extra cases, and control horses were used and when. Maybe a detailed table would solve this issue. The detailed comments are within the text.

Author Response

Reviewer 1

Dear Reviewer,

thank you for your thorough review of the manuscript and for the assessment of the study. We have implemented the suggestions you made for improving the manuscript point by point. The Materials and Methods section has been redesigned. Unnecessary subheadings that had made the structure very confusing have been deleted. Instead, the necessary information has been integrated into the Results and Discussion sections. Information on the horses studied is now summarized in a table, as suggested by you.

The results section has been structured more clearly. Most importantly, the tables are now smaller and clearer.

In addition, the introduction and discussion have been significantly revised.

Reviewer 2 Report

This manuscript presents the accumulation and metabolization of protoxins from sycamore maple (Acer pseudoplatanus) in various tissues from horses showing atypical myopathy cases. The experimental design was adequate to the objectives, and the study was well written.

Sapindaceae: Do not use italic letters for a family name.

In the Material and Methods section, I have some comments about the method validation. I did not understand the exact meaning of the coefficient of variation for tissue extraction. Is this procedure the trueness and precision (repeatability and within-lab reproducibility) or the recovery? Did you spike blank samples with the analytes at distinct levels? How many replicate samples did you use for determining the completeness of the extraction?  I did not understand how the stability of analytes was determined. The presentation of the method validation results should be improved.

Minor editing of English language required. For ex., metabolisation.

Author Response

Reviewer 2

Dear Reviewer,

thank you for your thorough review of the manuscript and for the assessment of the study. We have implemented the suggestions you made for improving the manuscript point by point. The Materials and Methods section has been redesigned. Unnecessary subheadings that had made the structure very confusing have been deleted. Instead, the necessary information has been integrated into the Results and Discussion sections. In fact, our unclear presentation of the coefficient of variation did not benefit the paper. We have deleted this part and have now addressed the problem in the discussion.

In addition Information on the horses studied is now summarized in a table and in the results section the tables are now smaller and clearer. Also, the introduction and discussion have been significantly revised.

Reviewer 3 Report

Overall thank you for providing this paper to Animals. There is a tremendous amount of work that went into the preparation of this manuscript. However, there are some significant points that protract the overall utility of this paper. 1. the introduction is way to long and while it is interesting that people can be poisoned by this toxin it really serves no proper addition to the paper to the extent that is included. 2. There is little direction to the introduction ie why is it important that we learn to test for these protoxins and toxins and what tissues should we consider. 3. I am not sure based on how this paper is laid out about your true hypothesis or null hypothesis and what are the research questions that you are trying to answer. Advice to significantly shorten the introduction and get to the point of what you are looking for and trying to answer. 

Materials and Methods: distinctly laid out however there is no discussion about what the horses were, how they were diagnosed, what were the ages, breed, sex etc. I am not really sure about the random comments about testing for coefficient of variation from 1 kidney 5 different times. Why just one kidney etc. 

There is absolutely no explanation for sample collection, handling, preparation and fixation etc. This seems to be a very fatal flaw to this paper because of tissue autolysis. 

Results: This section is way to busy, you have really long tables that drag from one page to the next and it just appears that you listed all raw data with zero explanation about statistical analysis but some must have been done due to the fact that you state means and standard deviations. There is no explanation of the data is normally distributed vs not normal distribution etc. You have not really explained at all what the results of the paper are you just list randomness on a page ie was anything significant or is this up to the reader to interpret. You need to state what your results are not just make tables.  The results section needs major revision to make it more concise. 

Discussion:

This section is far too long, you are introducing human information that really has no supportive evidence based on what you tested, it adds no substance and if anywhere needs to be in the introduction. The second paragraph of the discussion is probably one of the most important with statements that reflect your results. There are large sections about metabolism of the protoxins that really do not belong in the discussion. The purpose of the discussion is to explain the results but with your results so randomly described this makes the discussion very cumbersome to read and really make no logical sense. This needs removed and significantly rewritten or potentially moved to a completely different paper discussing the metabolism of the protoxin in the horse. You have so many abbreviations listed in the discussion with no explanation as to what they are for. This needs to be clearly stated at the beginning of the paper in a legend. You make statements that your values were within the control range but there are no control ranges listed. 

Overall the grammar and English used in this paper is not terrible. There are a few sentences that are cumbersome with multiple commas and weird structure that makes it slightly difficult to read. Overall this paper is much too long and there is too much included which in turn makes small sentence structure errors more evident. 

Author Response

Reviewer 3

Dear Reviewer,

thank you for your thorough review of the manuscript and for the assessment of the study. We have implemented the suggestions you made for improving the manuscript point by point.

The Introduction section has been shortened and redesigned overall. In its new form, the statements now better guide the reader toward the research question and, the objectives have now been clearly stated.

The missing information about the horses is now summarized in a table. Sample collection and handling are described in the materials section.

The results section has been structured more clearly. Most importantly, the tables are now smaller and clearer. For the acyl conjugates a normal distribution has not been indicated, because this is not known for the tissues. But the mean values found in measurements of our laboratory for materials of unaffected horses have been given in a section of Table 4. For Table 2, of course, a normal distribution does not exist because the toxins and toxin metabolites are not physiologically present in the tissues.

Round 2

Reviewer 1 Report

Dear Authors thank you for applying my comment and suggestions. I believe the manuscript is suitable for publication in its present form.

Author Response

Dear Reviewer,

thank you again for reviewing our article. Your comments were very helpful.

Kind regards

J. Sander

Reviewer 3 Report

There are still areas of this manuscript that warrant improvement. 

First, there is not statistical description of what you used for your data sets. You only use mean values which implies that the data is normally distributed. Was the data tested for normal distribution if so how and using what methodology. Please provide the methodology for all statistical methodologies used within the manuscript. 

Also, can you please define further if and why you state the horses were poisoned, this typically implies to an illicit nature or harm that is done to the animal rather than natural ingestion resultant from the ingestion of a toxin or in your case presumably protoxin that was metabolized to the active toxin resulting in the patient having an intoxication. 

The discussion still needs improvement do not start the discussion with "Here it was shown" consider revising the first paragraph of the discussion to read smoother. An example This paper is the first to describe......

Also, revise lines 267-271, this reads very awkward and implies first hand opinion. 

Overall the manuscript is much improved from the original but still requires further editing. 

Minor improvements required. 

Author Response

Thank you again for your comments. They really helped us to improve our manuscript.

You comment, that  there is no statistical description regarding our data sets. A normal distribution does, of course, not exist for the toxins and their metabolites, because normally the concentration is zero. For the acyl components, it would be possible to determine normal distributions in the tissues, but this would be far beyond the scope of the study. We have therefore contented ourselves with mean values from a few comparative analyses, but have now also highlighted this in the text.

You are critical of the term poisoning. We have replaced it as far as possible.

We have also taken account of the other comments